# FOLIAGEN: FRAMEWORK FOR FOLIAGE IMAGE GENERATION FROM INDIVIDUAL CROP LEAF IMAGES

## ABSTRACT

While machine learning (ML)-based crop disease classifiers mostly targeted individual leaf images, real-world applications call for disease classification on crop foliage images instead, because they usually rely on cameras mounted on unmanned aerial vehicles to capture foliage images across vast crop fields for automated disease identification. We found that known state-of-the-art (SOTA) classifiers on the only real-world soybean foliage image dataset all exhibited unsatisfactory performance, despite the dataset being modest-sized and including just two soybean disease categories (among many). Hence, it is desirable to make available large foliage image datasets with common crop disease categories for better evaluating and possibly improving SOTA crop disease classifiers on foliage images. This paper introduces a framework that generates crop foliage images utilizing available datasets of individual leaf images, termed Foliagen (short for foliage generation). A generated foliage image dataset can be arbitrarily sized, with each image emulating the natural distribution of diseased leaves with a specified disease rate. Being annotated by design, such generated datasets are valuable for (1) evaluating the SOTA classifiers when applied to practical use and (2) pre-training general SOTA classifiers, making it possible to effectively fine-tune them using any real-world foliage image dataset for improved classification performance. The Foliagen framework is exemplified by generating foliage image datasets for soybean and tomato. Our evaluation results indicate that five SOTA classifiers on generated datasets with nine disease categories achieve accuracy up to 87% for soybean and 86% for tomato under $\gamma = 5\%$, and that they all exhibit less than 92% in classifying the real soybean foliage image dataset (with just two disease categories). Foliagen makes it possible to generate crop foliage image datasets to evaluate future disease classifiers objectively, aiming at in-field applications.

## 1 INTRODUCTION

Crops have been indispensable to human civilization since its inception, serving as a fundamental source of food, medicine, clothing, shelter, and oxygen. Extensive pursuits in crop's structure, phyllotaxis phenomenon (Coussement et al., 2018; Koki et al., 1994; Niklas, 1988), life cycle, and disease have been undertaken, aiming at yield improvement to meet growing demands. According to the Food and Agriculture Organization (FAO), crop diseases cost approximately $220 billion annually in the world (cro, 2021), with soybean alone accounting for a loss up to $3.9 billion USD in the USA (Bradley et al., 2021). They usually show prominent symptoms, manifesting themselves as changes in the soybean's leaf foliar appearance and/or shape. For example, rust in soybean exhibits small, pale green to yellow spots on the upper surface of leaves (see Figure 1(f)). Since many crop disease categories possibly exist as illustrated in Figures 1 and 2, early disease identification makes it possible to apply proper measures at the onset of diseases for curbing damage they may cause, retaining crop yields as best as possible.

Instead of relying on experienced farmers for disease identification, machine learning (ML) has been adopted (Abbas et al., 2021; Karlekar and Seal, 2020; Pan et al., 2023; Sun et al., 2024a;b; Wu et al., 2023; Yogabalajee et al., 2024) recently to classify soybean and tomato diseased leaf images with success. ML models automate disease classification through training on large amounts of diseased and healthy leaf images. As exemplified in Figures 1 and 2, quality images of individual diseased soybean and tomato leaves exist in public datasets (Bevers et al., 2022; Hughes et al.,

2015; Sivm205, 2023) for model training. A state-of-the-art (SOTA) ML model targeting soybean disease classification is shown to have an average accuracy above 85% on a non-public dataset of individual leaf images gathered from the soybean plantations of the author's college (Wu et al., 2023). Meanwhile, two SOTA ML models for tomato disease classification are demonstrated to enjoy high accuracy rates, with one under two public datasets of individual leaf images (given in (Gehlot et al., 2023; Hughes et al., 2015)) to exceed 99% (Sun et al., 2024b). Note that those SOTA classifiers often resort to specific augmentation strategies; for example, random masking on the individual soybean leaf images (Wu et al., 2023) or the Gaussian filter for enhancing and obscuring artifacts of individual tomato leaf images (Sun et al., 2024b). Such augmentation strategies can be expensive and less effective when applied to images with large numbers of leaves, like crop foliage images (see Figure 5 and Figures 6 and 7).

Although SOTA ML models (e.g., (Pan et al., 2023; Sun et al., 2024b; Wu et al., 2023; Yoga-balajee et al., 2024)) are high classification performers on the existing datasets of individual leaf images, they lack practicality, for in-field applications, where sequences of foliage images are usually captured over areas of interest by the cameras of unmanned aerial vehicles (UAVs) for disease identification. Therefore, accurate automated disease classification on foliage images is essential for practical applications. While one early work (Tetila et al., 2017) dealt with a small collection of foliage images covering only two diseased categories, it required highly experienced agronomists with substantial time and effort to annotate segmented parts of foliage images for accurate classification, considered too expensive and impractical to apply for in-field applications where large volumes of images are involved. So far, there is just one annotated soybean foliage image dataset available to the public, MH-SoyaHealthVision (Shinde and Attar, 2024), and no publicly available dataset of tomato foliage images exists, to the best of our knowledge. Unfortunately, the MH-SoyaHealthVision dataset fails to include many predominant soybean disease categories (see Figure 5 and Figures 6 and 7 in Appendix) and is likely to be inadequate to train an effective classifier for identifying the diseases at their early stages, since its images were not captured at the disease onset and thus often had considerable diseased leaves each.

This paper pioneers a framework to generate crop foliage images utilizing available datasets of single leaf images, called Foliagen (short for foliage generation). A generated foliage image dataset can be arbitrarily sized, with each of its images having a specified rate of diseased leaves (denoted by $\gamma$) and the rest being healthy. Such generated datasets are annotated by design, tailored to in-field applications for foliage image classification. They are valuable for (1) evaluating the SOTA classifiers when applied to practical use and (2) pre-training general SOTA classifiers, making it possible to fine-tune them using any real-world foliage image dataset for improved classification performance. The Foliagen framework is exemplified by generating foliage image datasets for soybean and tomato, using the individual diseased leaf images from ASDID (Bevers et al., 2022), PlantVillage (Hughes et al., 2015), and Kaggle (Sivm205, 2023) datasets. It takes as its input (1) the disease category and (2) the rate ($\gamma$) of diseased leaves in a foliage image. Each generated foliage image dataset covers nine predominant disease categories for soybean and tomato (with samples depicted in Figure 5 and Figures 6 and 7 in the Appendix, respectively). Such datasets make it possible to train classifiers for early disease identification when generating foliage images at a small $\gamma$ (say, 5%).

Unlike an individual leaf image where background or noise takes up its considerable area, a foliage image synthesized by Foliagen is dominated by soybean or tomato leaves, with only a small fraction of its area being background (see Figure 5 and Figures 6 and 7 in Appendix). It is found from our evaluation results that SOTA models classify nine disease categories less accurately on the generated soybean foliage image datasets than on the original individual leaf images, at varying degrees. As listed in Table 1 for soybean disease classification, SOTA classifiers are subject to accuracy reduction by up to 3% (or 12%) on generated foliage image datasets with $\gamma = 15\%$ (or 5%), rendering the best performer (VGG19 (Simonyan and Zisserman, 2015)) for classifying individual soybean images to be less attractive for foliage image classification. Similar performance degradation is observed for tomato disease classification, up to 17% (or 22%) reduction in accuracy for Swin Transformer (Liu et al., 2021) under generated foliage image datasets with $\gamma = 15\%$ (or 5%), as shown in Table 3. The most effective classifier under foliage image datasets is DenseNet121 (Huang et al., 2017), instead of VGG19 (Simonyan and Zisserman, 2015) on the datasets of individual tomato leaf images. Hence, Foliagen establishes foliage image datasets useful for candidly assessing known and future classifiers to identify ones that are most effective for in-field applications.

In addition, classifiers pre-trained by a generated soybean foliage image dataset with $\gamma = 15\%$ is confirmed to perform equally well (with VGG19 to achieve $93^+\%$ accuracy) under the only known real-world foliage image dataset (e.g., MH-SoyaHealthVision (Shinde and Attar, 2024)) via employing a small fraction of the dataset (say, 10%) to fine-tune the pre-trained classifiers, as the result of transfer learning. Classifiers so pre-trained learns abstract features of diseased soybean foliage images under various disease categories, making it possible to adapt soundly for classifying the real-world foliage images with high accuracy. The overall contributions of this paper are summarized as follows:

- A framework for generating annotated foliage image datasets (Foliagen) is introduced by utilizing public datasets of individual crop leaf images. Datasets so generated can be arbitrarily large, properly annotated, and aimed to cover various crop disease categories common in the field and to target early disease identification by setting a small $\gamma$ (say, 5%).

- Foliagen is exemplified by generating foliage image datasets for soybean and tomato, with the generated datasets used for evaluating SOTA classifiers to determine the most effective ones for real-world applications.

- We have demonstrated that generated foliage image datasets can pre-train a general model for crop disease classification, so that the pre-trained model can then be fine-tuned by a small fraction of any real-world foliage image dataset for high classification accuracy under the dataset, as a result of transfer learning.

## 2 RELATED WORK

### 2.1 PLANT LEAF DATASETS

Many plant leaf datasets are available to the public, with some covering a variety of plant species each, such as PlantVillage (Hughes et al., 2015) and PlantDoc (Singh et al., 2020), and others being plant-specific, including those for soybean and tomato given in (Bevers et al., 2022; Gehlot et al., 2023; Hughes et al., 2015; Shinde and Attar, 2024; Sivm205, 2023). Existing plant leaf datasets are outlined briefly below, with more details about the PlantVillage dataset (Hughes et al., 2015), ASDID (Bevers et al., 2022), a Kaggle dataset (Sivm205, 2023), and the MH-SoyaHealthVision dataset (Shinde and Attar, 2024) provided in Section 3.1.

**PlantVillage dataset.** PlantVillage (Hughes et al., 2015) contains over 54,300 expertly curated healthy and diseased leaf images from various plants, including thirteen major crop species like soybean, tomato, etc. Its leaf classification has been attempted by GoogleNet (Szegedy et al., 2015) and AlexNet (Krizhevsky et al., 2012) to attain high accuracy.

**PlantDoc dataset.** The PlantDoc dataset (Singh et al., 2020) contains 2,598 single-leaf images of 17 disease categories across 13 plant species, including tomato and soybean. With its images gathered in a controlled laboratory environment, this dataset has limited applicability in real-world scenarios.

**ASDID.** Auburn Soybean Diseased Image Dataset (ASDID) (Bevers et al., 2022) provides high-quality individual leaf images of the soybean plant, covering 9 disease categories, namely, bacterial blight, cercospora leaf blight, downy mildew, frogeye leaf spot, soybean rust, target spot, and potassium deficiency. The dataset was captured primarily at the EV Smith Agricultural Research Station in Tallassee, Alabama, and added with 80 images per disease category from the publicly available Image Database of Plant Disease Symptoms (PDDB) (Barbedo et al., 2016). Several ML-based classifiers (He et al., 2016; Huang et al., 2017; Simonyan and Zisserman, 2015) were employed to evaluate the dataset, with DenseNet201 (Huang et al., 2017) achieving the highest performance.

**Kaggle dataset.** Soybean Diseased Leaf Dataset contains individual leaf images from Kaggle (Sivm205, 2023), embracing 10 disease categories and having an artificially generated complex background added to each image. Among the 10 diseases, only mosaic virus and sudden death syndrome are considered.

**FieldPlant dataset.** FieldPlant (Moupojou et al., 2023) is a dataset of individual crop leaf images annotated by pathologists, containing 8,629 images across 27 disease categories for three crops, including tomato. It aims at practical crop disease classification with every leaf image involving a complex background.

**Tomato-Village dataset.** Compensating for the negative effects due to the laboratory-controlled environment setup for gathering PlantVillage's leaf images, the Tomato-Village dataset (Gehlot et al., 2023) contains real-world images, which belong to three groups, respectively for (1) multi-class tomato disease classification, (2) multi-label tomato disease classification, and (3) object detection-based tomato disease detection. This dataset covers seven disease categories, namely, early blight, late blight, leaf miner, magnesium deficiency, nitrogen deficiency, potassium deficiency, and spotted wilt virus, for multi-class classification applications.

**MH-SoyaHealthVision dataset.** As far as we know, MH-SoyaHealthVision (Shinde and Attar, 2024) is the only public and well-annotated dataset of diseased soybean foliage images. The dataset provides both ground-level leaf images and foliage leaf images, collected using a UAV, from the soybean fields of Maharashtra, India. It comprises a total of 5,680 high-resolution images grouped into (1) single leaf images of 4 diseases categories (i.e., frogeye, mosaic virus, septoria brown spot, and rust), two types of pest attacks, and healthy leaves and (2) UAV-captured soybean foliage images, which belong to two disease categories of mosaic virus and rust, plus the healthy category.

All the above datasets were reviewed and carefully examined for possible use by our Foliagen framework; however, only the PlantVillage, ASDID, and the Kaggle datasets were selected as single-leaf image sources due to their large numbers of well-annotated images, with high fidelity and clarity.

## 2.2 GENERATION METHODS

Synthetic dataset generation has been widely adopted in the field of computer vision, especially in areas such as disease detection, object detection, and segmentation, where data collection is very costly and time-consuming. Various methods have been experimented with in previous research to augment image data, including simple copy & paste, graphical method (Bradley et al., 2013), and machine learning techniques. Simple copy & paste has been utilized to augment data (Guo, 2024; Higuchi et al., 2023), and found to be effective for object detection (Dwibedi et al., 2017), image classification (Mesnage et al., 2025), and instance segmentation (Ghiasi et al., 2021; Remez et al., 2018; Shen and Li, 2023). (Dwibedi et al., 2017) introduced the concept of cut and paste to augment image data for instance detection, demonstrated to yield marked improvements under the LVIS benchmark (Ghiasi et al., 2021)and improved training performance for ultrasound instance segmentation (Shen and Li, 2023).

Deep learning (DL) methods were also implemented to augment leaf data, such as LeafNST (Khare et al., 2024), NeuraLeaf (Yang et al., 2025), LeafGAN (Cap et al., 2020), and others (Benfenati et al., 2022; Ward et al., 2018). Particularly, (Benfenati et al., 2022) adopted a Residual Variational Autoencoder for leaf generation and a generative adversarial network, Pix2pix, for color translation on generated leaf images. Similarly, LeafNST Khare et al. (2024) transfers the symptoms of diseased leaves into healthy leaves to enlarge the diseased leaf count. While these DL models are proven to be effective for data augmentation, they are limited to single leaf images, inevitably constraining their usage in the real world.

## 2.3 CROP LEAF DISEASE CLASSIFICATION MODELS

Automated disease classification studies have been conducted lately based on the aforementioned publicly available datasets, plus certain privately collected datasets, to exhibit impressive classification outcomes (Abbas et al., 2021; Bouni et al., 2024; Karlekar and Seal, 2020; Pan et al., 2023; Sun et al., 2024a;b; Wu et al., 2023; Yogabalajee et al., 2024; Yu et al., 2022). Bevers *et al.* (Bevers et al., 2022) in Auburn, AL collected a high-quality single leaf imagery dataset and employed standard CNN-based models, such as VGG19, DenseNet201, and ResNet50, etc., to classify their collected dataset, achieving high accuracy. Enhanced DenseNet121 (Yogabalajee et al., 2024) relied on transfer learning to tackle the intricate challenges of classifying individual soybean leaf images. Using a conditional generative adversarial network (C-GAN) to generate synthetic individual tomato diseased leaves, previous work (Abbas et al., 2021) achieved very high performance employing DenseNet121 (Huang et al., 2017) as its classification model. Merging classical feature engineering with modern machine learning techniques, Bouni *et al.* (Bouni et al., 2024) recently have employed a CNN pre-trained on ImageNet (Deng et al., 2009) under mutual information-based feature fusion to get a high performer for tomato disease classification.

While CNN-based models can capture the prominent features of images, incorporating an attention mechanism into the models enables them to emphasize the region that contributes most to performance improvement. Previous studies incorporating attention mechanisms (Hu et al., 2018; Liu et al., 2021; Sun et al., 2024a;b; Wang et al., 2020; Woo et al., 2018; Wu et al., 2023) have been proven effective for computer vision tasks, such as image segmentation, image classification, and object detection targeting leaf disease classification, where leaf colors and shapes are key features of interest (Pan et al., 2023; Wu et al., 2023). Swin Transformer (Liu et al., 2021) introduces a hierarchical segmentation along with the vision transformer to implement shifted windows, which capture main features across different segmentation regions of an image. Meanwhile, CBAM-ConvNeXt (Wu et al., 2023) employs both channel attention and spatial attention plus ConvNeXt (Liu et al., 2022), to classify the individual soybean leaf images, which are not publicly available yet. While exhibiting impressive feats, all known studies (but (Tetila et al., 2017)) aim only at classifying high-resolution images of individual crop leaves, as illustrated in Figures 1 and 2.

## 3 METHODOLOGY

The Foliagen framework generates diseased foliage images out of available single-leaf diseased images for the classification of diseased foliage images for (1) objectively evaluating known and future crop disease classifiers when deployed for in-field applications and (2) pre-training general crop disease classifiers, making them tailored for specific fields with high classification performance after fine-tuned by a small number of annotated foliage images gathered in those fields, as the result of transfer learning. It is exemplified by generating foliage images of soybean and tomato, leveraging three publicly available individual-leaf image datasets, ASDID Bevers et al. (2022), a Kaggle dataset Sivm205 (2023), and the PlantVillage dataset (Hughes et al., 2015), which are detailed in Section 3.1. Single soybean leaf images are preprocessed (as described in Section 3.1) before being utilized by Foliagen to generate diverse sets of foliage images for evaluating known classifiers objectively. Such a single image manipulation methods has been proven to be effective for image segmentation and object detection data augmentation (Dwibedi et al., 2017; Ghiasi et al., 2021; Remez et al., 2018). In addition, known classifiers after being pre-trained by generated foliage images that cover 9 disease categories are fine-tuned via 10% of images from the real-world soybean foliage image dataset, MH-SoyaHealthVision Shinde and Attar (2024), and are found to classify the remaining 80% MH-SoyaHealthVision images with improved performance.

### 3.1 DATA COLLECTION

**Soybean.** A total of 10,722 high-resolution single leaf images covering 7 disease categories from ASDID Bevers et al. (2022) and 132 images (with 22 for Mosaic Virus and 110 for sudden Death Syndrome) from a Kaggle dataset (Sivm205, 2023) were chosen for foliage image generation. Images of those nine disease categories (as depicted in Figure 1) feature diverse and complex natural/artificial backgrounds, which are undesirable when generating foliage images. Therefore, we pre-process the images to remove their undesired backgrounds (i.e., to make them transparent) using an open-sourced and AI-enabled background remover, rembg (Gatis, 2021), as shown in the first stage of Overall Foliagen depicted in Figure 3. Freely available soybean field soil images are then included in generated foliage images as their backgrounds, to emulate the natural habitat of soybean plants as best as possible.

The only publicly available and properly annotated dataset of diseased foliage images, MH-SoyaHealthVision (Shinde and Attar, 2024), comprises two disease categories: soybean rust and mosaic virus. The original images have a very high resolution of $3840 \times 2160$ and the dataset is imbalanced, with rust images outnumbering healthy images by a factor of four, leading to biased predictions favoring the majority class and consequently reducing the model's generalizability. To address these issues, we crop the high-resolution foliage images into ones with a lower resolution to ensure a more balanced data distribution and to obtain a total of 3210 images, comprising 1084 rust images, 1027 healthy images, and 1099 mosaic virus images, respectively.

**Tomato.** The single-leaf diseased images of tomato are from the PlantVillage dataset (Hughes et al., 2015), with 9 primary tomato disease categories, and they are taken in a laboratory environment and with the dimension of $256 \times 256$, as shown in Figure 2.

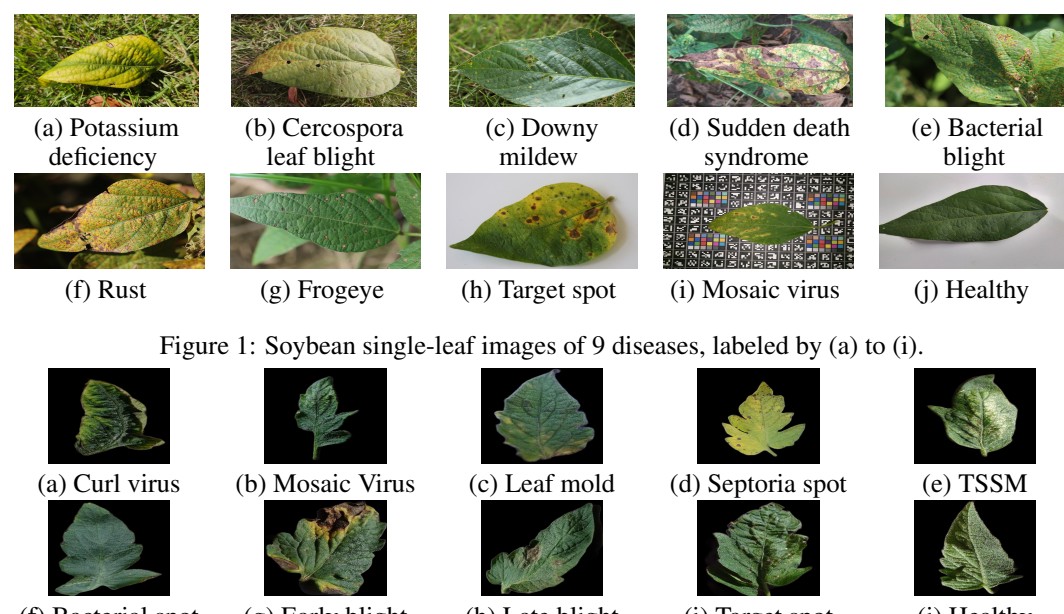

Figure 1: Soybean single-leaf images of 9 diseases, labeled by (a) to (i).

(a) Curl virus  (b) Mosaic Virus  (c) Leaf mold  (d) Septoria spot  (e) TSSM

(f) Bacterial spot  (g) Early blight  (h) Late blight  (i) Target spot  (j) Healthy

Figure 2: Tomato single leaf images of 9 diseases, labeled by (a) to (i).

## 3.2 FOLIAGEN FRAMWORK

Foliagen takes pre-processed individual leaf images as input to generate foliage images with a customizable rate of diseased leaves. After preprocessing single leaf images for background removal via rembg (Gatis, 2021), it then involves 3 levels of generation, Leaf Level, Plant Level, and Foliage Level, as depicted in Figure 3. Each level incorporates plant-specific information such as disease categories, target disease rate, and inherent leaf structural characteristics, as denoted by 'Configuration File' in Figure 3. The structure of the configuration file used to provide this information is detailed in Appendix A.6.

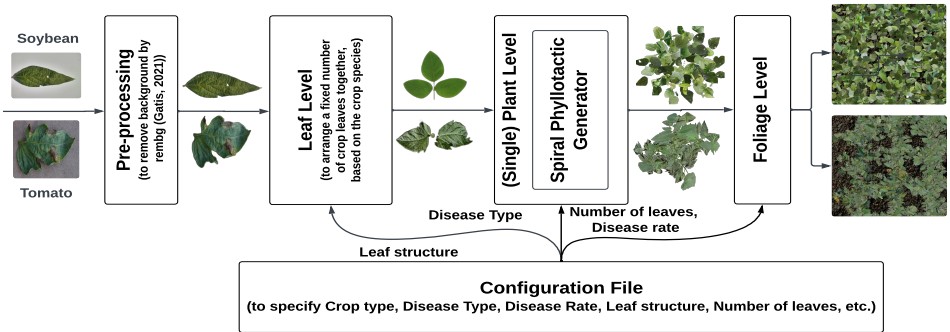

Figure 3: Overall Foliagen.

**Leaf Level.** Soybean leaves arrange themselves into 3 leaflets in each petiole, with a slightly bigger central leaflet and two lateral leaflets on the two side of the central leaflet, as illustrated in Figure 3. On the other hand, tomato leaves arrange themselves around a central axis, called the rachis. In practice, crop leaflets may or may not be diseased. Hence, a random number of healthy leaflets are included in each foliage image, governed by the disease rate $\gamma$.

**Plant Level.** The number of leaves in an adult crop differs largely, based on the crop. An adult soybean plant might contain 30-40 trifoliates, while an adult tomato plant may have 20-40 leaves, assuming the determinate variety of tomato crop. Both crops exhibit spiral phyllotaxis (Koki et al., 1994; Niklas, 1988), in which leaves are arranged in a spiral arrangement that makes the golden angle, i.e, 137.5°, to maximize sunlight exposure and minimize leaf overlap. Most of the leaves in

a crop are healthy in the early stages of a disease, so foliagen takes a small disease rate $\gamma$ (e.g., 5% or 15%). A spiral phyllotactic coordinates generator is the vital part of this Plant level, following the formulae given below to determine the coordinates and the angles of leaves to maintain spiral phyllotaxis (Niklas, 1988).

- Center: $(x_0, y_0)$,  Golden angle: $\theta_g^{rad} = 137.5° \times \frac{\pi}{180} = 1.57$,  Scaling factor: $s = 35$
- Number of trifoliates: $N = X \sim \mathcal{U}\{30, 31, \dots, 40\}$

For each trifoliate index $n \in \{1, 2, ..., N\}$:

$$\text{Angular displacement:} \quad \theta_n = n \cdot \theta_g^{\text{rad}} \qquad \text{Radial distance:} \quad r_n = s \cdot \sqrt{n}$$

$$\text{Cartesian coordinates:} \quad \begin{cases} x_n = x_0 + r_n \cos(\theta_n) \\ y_n = y_0 + r_n \sin(\theta_n) \end{cases}$$

The final discrete leaf positions are expressed by:

$$coords = \{(\lfloor x_n \rfloor, \lfloor y_n \rfloor) \mid n = 1, 2, ..., N\},$$

where $\lfloor \cdot \rfloor$ denotes the integer truncation.

As a tomato plant has branches, with a pair of leaves attached at a similar stem height and arranged in opposite directions, a sub-layer, called the branch layer, is added to create branches, each with 3-9 leaves. Such an emulated branch is then attached to the main stem in spiral phyllotactical order.

**Foliage Level.** Foliage images usually consist of multiple rows of crops planted in a farm field. Observation from real-world images taken using UAVs Shinde and Attar (2024) reveals that the major area of an image is covered by leaves, with only a small area being field soil (Freepik, 2025). As a result, Foliagen generates images with three rows of crops in each image to emulate their natural appearance. Samples of generated foliage images are illustrated in Figure 5 and Figures 6 and 7 in Appendix A.1.

### 3.3 DISEASE DISTRIBUTION

The distribution of disease in plant leaves is influenced by multiple interacting factors, including insect vectors, wind-mediated spore dispersal, plantation age, and environmental conditions, such as humidity, rainfall, and temperature. Although the spatial pattern of disease may vary considerably depending on these influences, a consistent phenomenon is that infections generally begin as localized hotspots on leaves or within a small plant patch and subsequently spread outward to one (or multiple) neighboring patch(es), ultimately forming a larger area of diseased foliage (Chen et al., 2025; Tao et al., 2021; Yang et al., 1991). This diseased leaf distribution is confirmed by us through examining the real-world foliage images of various disease categories in the available MH-SoyaHealthVision dataset (Shinde and Attar, 2024) (see Figure 4(c)), leading us to devise a three-level disease distribution pattern, as explained next.

**Region Level.** The whole image is divided into a $\alpha \times \beta$ grid with a total of $\alpha \times \beta$ regions. e.g., $4 \times 3$ = 12 regions. Most regions are disease-free when the disease rate ($\gamma$) is small, say $\leq 20\%$ and every diseased region is provided with a disease rate, so that all $\alpha \times \beta$ regions have the aggregate disease rate of $\gamma$. The diseased leaves in a diseased region are distributed normally across the region.

**Patch Level.** A patch refers to a collection of adjacent regions with one hotspot and its neighboring regions. Based on the disease rate, the number of diseased patches in a single foliage image varies from 1 to 2 under our disease rate of interest to be less than 20% for early disease identification. Naturally, a higher disease rate is expected to yield more disease patches in a patch. The distribution of the number of hotspots among the foliage images follows a skewed graph. Figure 4(b) shows the distribution of two hotspots for the mean disease rate of 15%, where the number of hotspot disease patches per image being 1 (or 2) equals 1,721 (or 6,279) out of 8,000 total generated foliage images.

**Dataset Level.** The disease rate of the foliage images is normally distributed with the standard deviation ($\sigma$) of 1.5 and the variable mean ($\mu = \gamma$) of 15% (or 5%). Figure 4(a) depicts the disease rate distribution for $\gamma = 15\%$, where the disease rate varies from 10% to 20%. For $\gamma = 5\%$, the disease rate ranges from 1% to 9%, with a similar normal distribution as illustrated in Figure 4(a).

**Crop Level Customization.** Foliagen is a common framework, aiming to generate a foliage imagery dataset for various crops out of those crops' single leaf images. Given crops differ among one

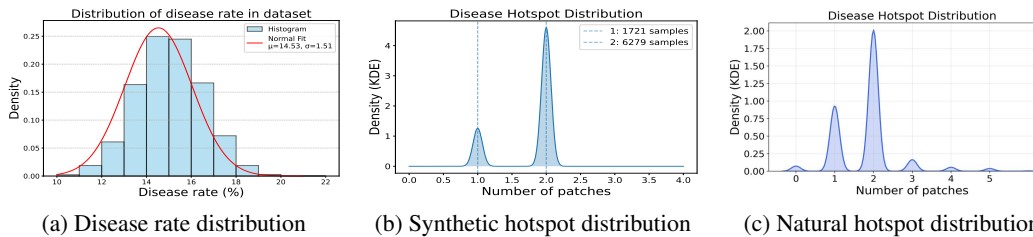

(a) Disease rate distribution     (b) Synthetic hotspot distribution     (c) Natural hotspot distribution

Figure 4: Disease distributions, (a) and (b) across synthetic diseased soybean foliage datasets ($\gamma$ = 15%), and (c) for the natural foliage dataset (MH-SoyaHealthVision).

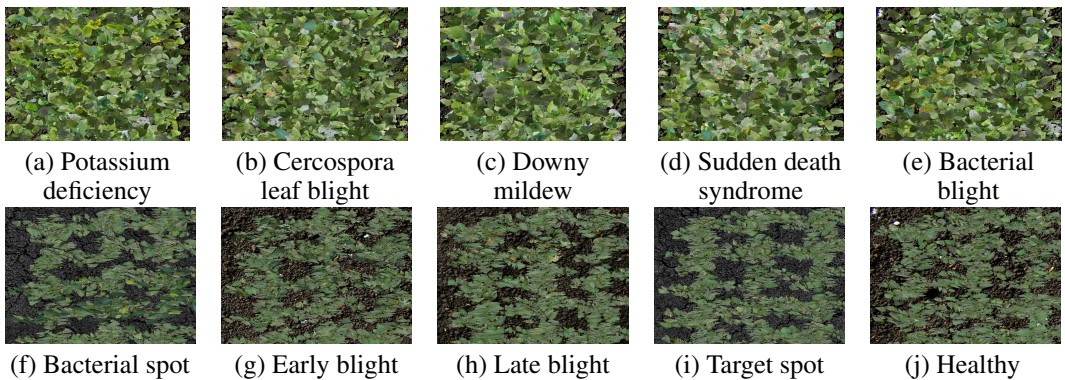

(a) Potassium deficiency    (b) Cercospora leaf blight    (c) Downy mildew    (d) Sudden death syndrome    (e) Bacterial blight

(f) Bacterial spot    (g) Early blight    (h) Late blight    (i) Target spot    (j) Healthy

Figure 5: Generated soybean foliage images (a)-(e) and tomato foliage images (f)-(j), for $\gamma$ = 15%. (All generated foliage image categories for soybean and tomato are shown respectively in Figures 6 and 7 in Appendix A.1.)

another in many factors, such as the leaf shapes, the leaf arrangements, phyllotaxis, number of leaves in single branch, numbers of leaves in single branches, etc., Foliagen is provisioned with a configuration file as its input to account for the crops' variability, with the file listing such crop-specific customization parameters as the disease rate, the size of individual leaf, the size of individual plant, the foliage size, disease categories, etc., to properly emulate crops' natural structures.

## 4 EVALUATION AND RESULT DISCUSSION

Extensive experiments to evaluate the performance of SOTA crop disease classifiers on generated foliage images are conducted on two workstations, with one housing 2 NVIDIA GeForce RTX 3090 GPUs (each with 24 GB of GDDR6X VRAM) and another housing 2 NVIDIA RTX 6000 Ada GPUs (each with 48 GB of GDDR6X VRAM). The foliage image datasets of soybean and tomato for $\gamma$ = 5% and 15% (to emulate the early stage of disease) have been generated for evaluation, with the sensitivity results of classifiers to a wide range of disease rates provided in Appendix A.4. Each generated foliage image dataset contains about 800 images for every disease category, plus a similar number of healthy foliage images. The Adam optimizer was used for model training, since it is known to converge faster with better performance by dynamically adjusting the learning rate for each parameter based on the first and second moments of gradients. Each model was trained for a maximum of 100 epochs, with an early stopping mechanism (with patience of 5 epochs) to avoid local minima. The batch size was set to 4, restricted by the GPU memory limitation, and the learning rate was initialized to 0.000001.

### 4.1 EVALUATION ON GENERATED SOYBEAN FOLIAGE IMAGE DATASETS

Various generated soybean foliage image datasets under different $\gamma$ (the rate of diseased leaves in each foliage image) values have been produced by Foliagen to objectively evaluate the SOTA disease classification models of VGG19 (Simonyan and Zisserman, 2015), ResNet50 (He et al., 2016), DenseNet121 (Huang et al., 2017), Swin Transformer (Liu et al., 2021), and CBAM-ConvNeXt (Wu et al., 2023) under exactly the same set of foliage images without any classifier-specific data pre-processing or manipulation. Each produced foliage image dataset covers all crop disease categories

that exist in the original datasets of individual leaf images, with a small $\gamma$ (say, 5%) to indicate an early disease stage. The evaluation results shed light on choosing the best classifier among those SOTA models for real-world soybean applications, where disease identification is based on in-field images captured by cameras mounted on UAVs. The evaluation metric outcomes under a synthetic dataset with a larger $\gamma$ are expected to be higher because more diseased leaves exist in each foliage image, making disease classification easier. The comparative performance evaluation results are obtained for $\gamma$ ranging from 5% to 15% and beyond, and they are found to follow similar trends. For simplicity, only the results for $\gamma = 5\%$ to 15% are listed in Table 1. It is evident from the table that DenseNet121 prevails for both $\gamma$ values, in terms of all the metrics.

Table 1: Comparative performance evaluation results (in %) under generated soybean foliage image dataset with $\gamma = 15\%$ (or 5%)

| Models | Accuracy | | F1-score | | Precision | | Recall | |
|---|---|---|---|---|---|---|---|---|
| VGG19 (Simonyan and Zisserman, 2015) | 76.26 | (65.45) | 77.17 | (65.67) | 78.11 | (66.19) | 76.26 | (65.15) |
| ResNet50 (He et al., 2016) | 85.31 | (81.91) | 85.40 | (83.58) | 85.69 | (85.37) | 85.11 | (81.87) |
| DenseNet121 (Huang et al., 2017) | **94.47** | **(87.56)** | **95.45** | **(90.81)** | **96.45** | **(94.45)** | **94.48** | **(87.44)** |
| Swin Transformer (Liu et al., 2021) | 72.36 | (65.38) | 72.99 | (64.50) | 72.37 | (63.62) | 72.68 | (65.41) |
| CBAM-ConvNeXt (Wu et al., 2023) | 77.76 | (66.33) | 79.73 | (69.59) | 81.84 | (73.15) | 77.72 | (66.36) |

## 4.2 EVALUATION ON REAL SOYBEAN FOLIAGE IMAGE DATASET

**Baseline.** The MH-SoyaHealthVision dataset (Shinde and Attar, 2024) was split into 80% for training, 10% for validation, and 10% for evaluation, enabling an objective evaluation of the same five SOTA classifiers under exactly the same set of real-world foliage images without any data preprocessing or manipulation. From the comparative performance results summarized under Baseline of Table 2, it is found that Swin Transformer achieves the highest performance, with accuracy exceeding 91% across all four metrics, whereas other models have the accuracy values ranging from $85^+\%$ to $89^-\%$. The baseline results indicate that Swin Transformer is the top performer for real-world applications.

**Transfer Learning.** Generated foliage image datasets can pre-train crop disease classification models to get powerful disease classifiers suitable for general applications. After those five SOTA models are pre-trained by our generated foliage images to cover nine categories of predominant soybean diseases, they are expected to serve as general classifiers for effectively identifying any real-world dataset of soybean foliage images at hand by fine-tuning them using a small fraction of foliage images in the dataset, due to transfer learning. When classifiers are pre-trained by a generated foliage image dataset with a small $\gamma$ (say, 5%), they are geared for identifying soybean diseases at an early stage, especially useful for real-world field applications. To this end, the trained models are evaluated under the real-world MH-SoyaHealthVision dataset (Shinde and Attar, 2024), after being fine-tuned via 5% images in the dataset, with the evaluation results listed under Pre-trained in Table 2. Note that the results are obtained when 10% and 85% dataset images are for validation and evaluation, respectively, after 5% images are employed for fine-tuning. Comparing the obtained evaluation results shown in Table 2, we find that the trained models with fine-tuning elevate performance metric values noticeably, to exceed 92% in accuracy for DenseNet121, ResNet50, CBAM-ConvNeXt, and Swin Transformer under $\gamma = 15\%$. The performance results of pre-trained models are worse under $\gamma = 5\%$ than under $\gamma = 15\%$, as expected, since the former aimed to detect diseases in an early stage, known to be harder but more useful. They also signify that Swin Transformer is the most desirable for in-field applications, when aiming at early disease detection (under $\gamma = 5\%$).

Table 2: Comparative performance evaluation results (in %) under MH-SoyaHealthVision

| | Baseline | | | | Pre-trained with $\gamma = 15\%$ (or 5%) | | | |
|---|---|---|---|---|---|---|---|---|
| Models | Accuracy | F1-score | Precision | Recall | Accuracy | F1-score | Precision | Recall |
| VGG19 | 85.38 | 86.6 | 87.98 | 85.38 | 88.00 (84.03) | 88.22 (84.06) | 88.41 (84.08) | 88.03 (84.03) |
| ResNet50 | 88.31 | 89.69 | 91.11 | 88.31 | **95.83** (90.61) | **95.91** (90.86) | **95.98** (91.12) | **95.84** (90.60) |
| DenseNet121 | 87.5 | 87.95 | 88.37 | 87.5 | 94.40 **(94.47)** | 94.48 **(94.19)** | 94.55 **(94.29)** | 94.40 **(94.08)** |
| Swin Transformer | **91.48** | **91.47** | **91.46** | **91.48** | 92.32 (87.57) | 91.33 (88.52) | 90.39 (89.51) | 92.29 (87.56) |
| CBAM-ConvNeXt | 88.82 | 87.06 | 85.39 | 88.83 | 95.38 (86.76) | 95.37 (88.06) | 95.36 (89.42) | 95.38 (86.74) |

### 4.3 TOMATO

Foliagen synthesizes various disease datasets of tomato foliage images based on the PlantVillage dataset (Hughes et al., 2015) (publicly available datasets of single leaf images with 9 primary tomato disease categories), under a range of $\gamma$ for evaluating the SOTA classifiers. The evaluation metric outcomes under a generated dataset with a larger $\gamma$ are expected to be higher because more diseased leaves exist in each foliage image, making disease classification easier. Given that the comparative evaluation results are obtained for $\gamma$ ranging from 5% to 15% and beyond follow similar trends, Table 3 lists only the results for $\gamma = 5\%$ and 15%. As evident from the table, the considered models all perform better under 15% than under 5% with respect to the four performance metrics, underscoring the fact that they tend to struggle in early disease detection (under $\gamma = 5\%$), especially for VGG19, ResNet50, and Swin Transformer. The evaluation results imply that DenseNet121 outperforms the rest consistently, making it the most desirable classifier for in-field applications for identifying tomato diseases according to foliage images captured in the field by UAVs.

Table 3: Comparative performance evaluation results (in %) under generated tomato foliage image dataset with $\gamma = 15\%$ (or 5%)

| Models | Accuracy | | F1-score | | Precision | | Recall | |
|---|---|---|---|---|---|---|---|---|
| VGG19 (Simonyan and Zisserman, 2015) | 87.66 | (80.53) | 87.71 | (81.01) | 87.64 | (81.51) | 87.78 | (80.50) |
| ResNet50 (He et al., 2016) | 92.22 | (78.84) | 92.44 | (81.39) | 92.65 | (84.11) | 92.23 | (78.83) |
| DenseNet121 (Huang et al., 2017) | **96.38** | **(86.93)** | **97.34** | **(89.47)** | **97.69** | **(92.16)** | **96.99** | **(86.93)** |
| Swin Transformer (Liu et al., 2021) | 79.80 | (66.53) | 79.36 | (66.71) | 78.84 | (66.81) | 79.88 | (66.60) |
| CBAM-ConvNeXt (Wu et al., 2023) | 81.91 | (64.85) | 81.68 | (65.56) | 82.96 | (66.24) | 81.96 | (64.89) |

## 5 CONCLUSION

This article introduces a framework (called Foliagen) to generate rich and arbitrarily-sized datasets of crop foliage images to cover all disease categories that exist in publicly available datasets of individual leaf images, with a given rate of diseased leaves ($\gamma$) in each foliage image generated to emulate the real foliage images captured in farm fields when their crop diseases are at the stage corresponding to $\gamma$. The generated foliage datasets are employed to better and objectively evaluate state-of-the-art leaf disease classifiers without invoking classifier-specific data pre-processing or manipulation. The evaluation results make it possible to choose the most effective crop classifier among SOTA ones for in-field applications with UAV-captured images (rather than individual leaf images) for disease identification. Being a generated foliage dataset, its primary limitation lies in the lack of naturalness and limited real-world applicability; however, the strong performance of crop disease classifiers pre-trained on it suggests its potential viability for broader applications. With an available in-field foliage dataset, the pre-trained models can be fine-tuned using a small fraction of the dataset images to yield effective disease classifiers targeting the field where the foliage dataset is gathered. While Foliagen is exemplified for classifying soybean and tomato diseases via SOTA models in this paper, it is readily useful for other crops and for objectively evaluating future disease classifiers aiming at in-field applications.

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

## A APPENDIX

This section contains additional evidence to support our dataset and data generation method.

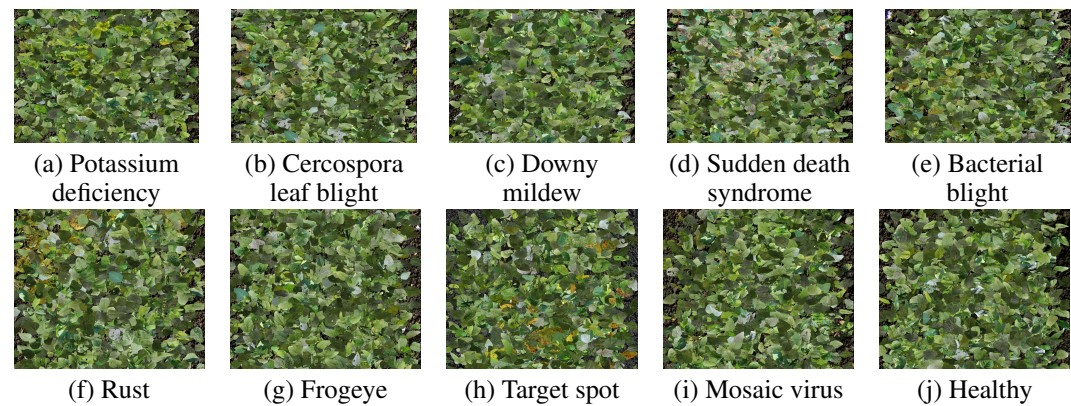

(a) Potassium deficiency    (b) Cercospora leaf blight    (c) Downy mildew    (d) Sudden death syndrome    (e) Bacterial blight

(f) Rust    (g) Frogeye    (h) Target spot    (i) Mosaic virus    (j) Healthy

Figure 6: Generated soybean foliage images of 9 diseases, labeled by (a) to (i), for $\gamma = 15\%$.

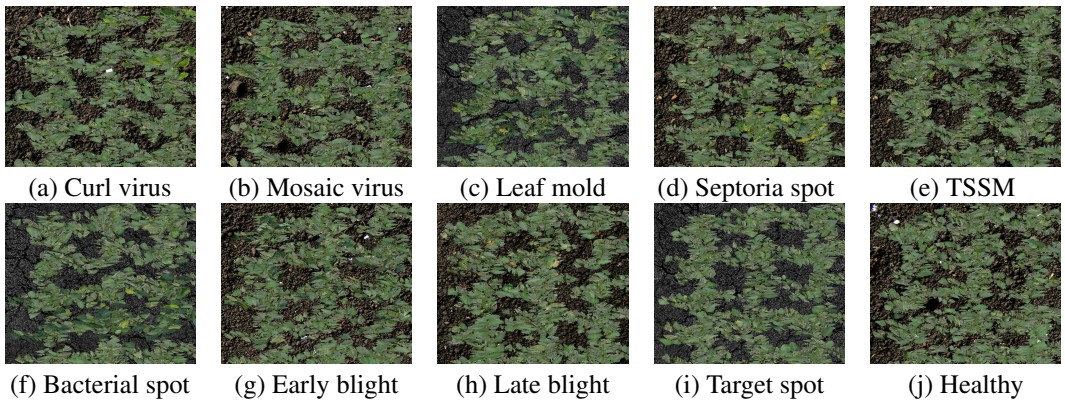

(a) Curl virus    (b) Mosaic virus    (c) Leaf mold    (d) Septoria spot    (e) TSSM

(f) Bacterial spot    (g) Early blight    (h) Late blight    (i) Target spot    (j) Healthy

Figure 7: Generated tomato foliage images of 9 diseases, labeled by (a) to (i), for $\gamma = 15\%$.

## A.1 GENERATED FOLIAGE IMAGES

The generated diseased foliage images for both Soybean and Tomato are shown next.

## A.2 NATURAL FOLIAGE IMAGE

Foliagen is based on the observation of natural foliage and the findings given in published articles, to obtain high quality data for foliage disease classification. Our work evaluates MH-SoyaVisionHealth (Shinde and Attar, 2024), a natural soybean foliage dataset with 2 disease categories, rust and mosaic virus. Figure 8 depicts real-world soybean foliage images collected using UAVs at a farm in India.

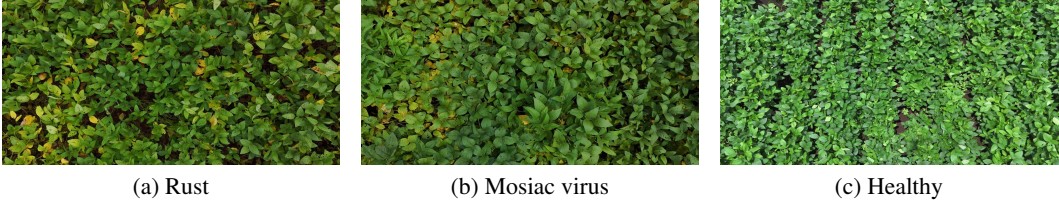

(a) Rust      (b) Mosiac virus      (c) Healthy

Figure 8: Sample soybean foliage images from the MH-SoyaVisionHealth dataset (Shinde and Attar, 2024).

### A.2.1 FOLIAGE GENERATION USING GENERATIVE MODEL

Recently, generative models have been used to generate almost all digital artifacts, including photos, videos, and texts. In the context of photos, the generative adversarial networks (GAN) and diffusion

models have been used extensively. Hence, we experiment with Deep Convolutional Generative Adversarial Network (DCGAN), Denoising Diffusion Probabilistic Model (DDPM), and PixelCNN to generate diseased foliage images from the real-world MH-SoyaVisionHealth dataset. All generative models are trained for 100 epochs before generating images from them, with their outcomes depicted in Figure 9.

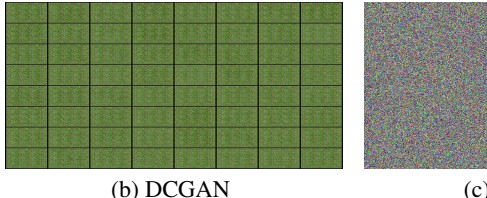

|          (a) DDPM          |          (b) DCGAN          |          (c) PixelCNN          |

Figure 9: Foliage generated using generative models.

## A.3    ADDITIONAL EXPERIMENTAL RESULTS

We conducted extensive experimentation with multiple strategies aimed at strengthening the viability and reliability of our dataset. These efforts included (1) exploring alternative data preparation and augmentation techniques and (2) comparing the results of Foliagen with the baselines to ensure that its resulting dataset captures realistic variability while preserving essential pathological characteristics. The outcomes of these investigations are presented in the following sections.

### A.3.1    TRANSFER LEARNING USING ASDID

The models were pre-trained on the raw ASDID dataset and then fine-tuned with 5% of the MH-SoyaVisionHealth dataset. Table 4 illustrates the transfer learning performance with all the hyper-parameters configured as discussed in Section 4.

Table 4: Comparative performance evaluation results (in %) under the MH-SoyaVisionHealth dataset, pre-trained on ASDID dataset

| Models | Accuracy | F1-score | Precision | Recall |
|---|---|---|---|---|
| VGG19 (Simonyan and Zisserman, 2015) | 33.33 | 32.56 | 33.33 | 32.94 |
| ResNet50 (He et al., 2016) | **74.84** | **75.31** | **75.80** | **74.82** |
| DenseNet121 (Huang et al., 2017) | 35.37 | 34.43 | 33.55 | 35.36 |
| Swin Transformer (Liu et al., 2021) | 69.45 | 69.92 | 70.38 | 69.45 |
| CBAM-ConvNeXt (Wu et al., 2023) | 49.32 | 50.25 | 51.22 | 49.32 |

### A.3.2    VARYING LEAF SIZES

The current version of the dataset reduces the size of individual leaves to a similar size as that of natural foliage leaves. This step gave a huge performance hike in the evaluated model. Many variations in leaf sizes, maintaining their aspect ratio, were used to create the dataset and were evaluated. One of the experimental results is shown in Table 5. It is vivid from the table that upscaling the leaf size leads to degraded performance for all classifiers; this is also true when the leaves are downscaled in size.

### A.3.3    COMPARISON OF MODEL METRICS

Foliagen, as demonstrated by the results in this paper, generates high-quality datasets for foliage disease classification in both soybean and tomato. Besides accuracy, computational efficiency is also important for consideration. Table 6 reports the model parameter count and the per-epoch training times when trained on a generated foliage dataset, compared with those of the baseline constructed from individual leaf images. As expected, single-leaf images incur lower training times per epoch due to their reduced visual complexity. Despite their substantially higher image complexity and larger spatial dimensions, foliage images are subject to only some 30% increases in per-epoch

Table 5: Comparative performance evaluation results (in %) under the generated tomato foliage image dataset with 2× upscaled leaf size

| Models | Accuracy | F1-score | Precision | Recall |
|---|---|---|---|---|
| VGG19 (Simonyan and Zisserman, 2015) | 65.23 | 65.41 | 65.59 | 65.23 |
| ResNet50 (He et al., 2016) | 76.73 | 77.61 | 78.51 | 76.73 |
| DenseNet121 (Huang et al., 2017) | **83.78** | **83.14** | **82.38** | **83.92** |
| Swin Transformer (Liu et al., 2021) | 67.95 | 68.96 | 70.02 | 67.91 |
| CBAM-ConvNeXt (Wu et al., 2023) | 71.88 | 71.96 | 72.03 | 71.88 |

training time, indicating that Foliagen produces results to train useful classifiers for real-world applications, with acceptable computational overheads.

Table 6: Comparative model evaluation metrics results under the generated dataset and the single-leaf image datasets of Soybean

| | For generated dataset ($\gamma$ = 15%) | | For single leaf images | |
|---|---|---|---|---|
| Models | No. of Parameters | Per epoch training time | No. of Parameters | Per epoch training time |
| VGG19 (Simonyan and Zisserman, 2015) | 139,611,210 | 459.17 | 139,611,210 | 331.03 |
| ResNet50 (He et al., 2016) | 23,581,642 | 433.38 | 23,581,642 | 244.10 |
| DenseNet121 (Huang et al., 2017) | 7,047,754 | 454.73 | 7,047,754 | 645.819 |
| Swin Transformer (Liu et al., 2021) | 27,527,044 | 419.10 | 27,527,044 | 290.10 |
| CBAM-ConvNeXt (Liu et al., 2021) | 29,727,934 | 437.34 | 29,727,934 | 331.03 |

## A.4 SENSITIVITY ANALYSIS OF DISEASE RATE

Figure 10 illustrates the sensitivity analytic results across disease rates, revealing consistent performance patterns among the classifiers. Accuracy increases for all models as disease prevalence intensifies, with DenseNet maintaining the highest and most stable performance levels across the full disease rate range and ResNet showing a similarly smooth upward trend. VGG improves more gradually, becoming competitive only at higher disease rates, whereas CBAM remains highly sensitive to disease severity, performing poorly at low levels but rising sharply once pathological cues become pronounced. Swin consistently yields the weakest performance with limited benefit from increased disease information.

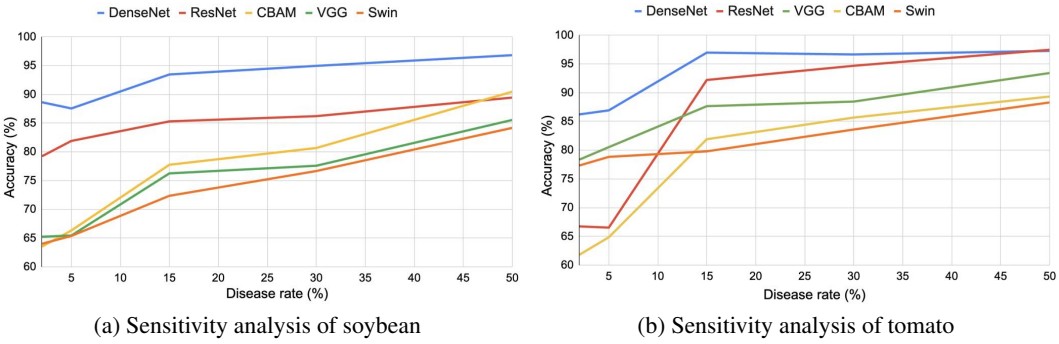

(a) Sensitivity analysis of soybean    (b) Sensitivity analysis of tomato

Figure 10: Sensitivity analysis on the disease rate for soybean and tomato.

## A.5 ABLATION STUDY

To investigate the contribution of each component of the Foliagen framework, we conducted an extensive ablation study on both soybean and tomato plants. Table 7 reports the performance of different framework variants for both crops. Specifically, we evaluate two configurations: (1) without the plant-level component, where naturally structured single-plant foliage is generated solely by arranging leaves according to the spiral phyllotactic pattern, and (2) without removing background from individual leaf images. The results clearly demonstrate that the full Foliagen framework consistently outperforms all ablated variants across all evaluated models. As expected, retaining the

background in leaf images leads to a substantial performance drop, highlighting the importance of clean leaf segmentation for realistic foliage synthesis.

Table 7: Ablation study of Foliagen for Soybean and Tomato ($\gamma = 15\%$)

| | Soybean | | | Tomato | | |
|---|---|---|---|---|---|---|
| Models | Ours | Without plant level | with background | Ours | Without plant level | With background |
| VGG19 | **76.26** | 69.29 | 37.52 | **87.66** | 82.60 | 53.71 |
| ResNet50 | **85.31** | 82.01 | 45.78 | **92.22** | 87.39 | 55.32 |
| DenseNet121 | **91.47** | 84.66 | 76.66 | **96.98** | 90.32 | 71.63 |
| Swin Transformer | **72.36** | 65.94 | 32.58 | **79.80** | 66.53 | 43.84 |
| CBAM-ConvNeXt | **77.76** | 73.15 | 49.15 | **81.91** | 72.51 | 59.50 |

## A.6 CONFIGURATION FILE

As depicted in Figure 3, all level of foliage generation require plant specific information to generate high quality foliage images. The configuration file includes number of leaves, disease rate, and other factors listed next.

```
{

  "num_leaves": 50,
  "diseases": "list_of_diseases",
  "foliage_size": "(1024, 1500)",
  "single_plant_size": 512,
  "single_leaf_size": 70,
  "num_plants": 16,
  "plant_offset": 100,
  "disease_rate": 15,
  "background_image_path": "<path_to_background_images>",
  "input_path": "<path_to_raw_input_images>",
  "output_path": "<path_to_save_generated_images>",
  "type": "tomato",
  "leaf_spacing": 60
}
```

