# OpenReview forum: "Foliagen: Framework for Foliage Image Generation from Individual Crop Leaf Images"
_ICLR.cc/2026/Conference — Submitted to ICLR 2026_

### Official Review · Reviewer_4gCG · 2025-10-30

**Soundness:** 2
**Presentation:** 3
**Contribution:** 2
**Rating:** 6
**Confidence:** 4

**Summary:**

This paper introduces Foliagen, a framework for synthesizing crop foliage images from single-leaf datasets by incorporating biologically-informed rules (like phyllotaxis and disease hot spots). Pre-training on this synthetic data and fine-tuning on a small real dataset significantly enhances performance, bridging a critical gap between lab-trained models and real-world UAV applications.

**Strengths:**

1.  Significance and Practical Impact: The work addresses a crucial, real-world data gap (scarcity of real crop canopy images) and offers a valuable, cost-effective solution through synthetic data pre-training.
2.  Biologically-Informed Approach and Ablation: The use of biological rules for structure and disease placement is a key feature. Crucially, experiments (Appendix A.2.1) justify the framework's necessity by showing its superiority over direct single-leaf pre-training.

**Weaknesses:**

1.  Overly Simplified and Non-Generalizable Framework Diagram (Fig. 3): The main diagram is a simple linear flow that fails to illustrate core algorithmic details. More importantly, it is *Soybean-specific* (showing trifoliate synthesis) and completely omits the process for the second example crop, tomato, undermining the claim of a general "framework."
2.  Lack of Simple Synthesis Baseline Comparison: A key ablation is missing: a comparison against a much simpler baseline, such as **random copy-pasting** of leaves. Without this, it is unclear if the complex biological modeling (e.g., phyllotaxis) is truly necessary for effective transfer learning.
3.  Ambiguity and Errors in Core Mathematical Formulation (Sec. 3.2): The formulas for leaf positioning contain two issues: (a) The golden angle definition ($\theta_{g}=137.5^{\circ}$) conflicts with the use of the non-standard $\theta_{g}^{rad}$ in the cosine/sine functions, creating a unit ambiguity. (b) An "off-by-one" error exists in the index range $n\in\{1,2,...,N-1\}$, which does not match the intended leaf count $N$.
4.  Lack of Real-World Distribution Validation for Disease Modeling: The paper claims its disease modeling (Fig. 4) is based on real data, but only presents statistics for the *synthesized* dataset. To justify fidelity, the authors must first present an analysis of the **real-world target dataset (MH-SoyaHealth Vision)** to show that the distribution characteristics are faithfully replicated.

**Questions:**

please refer to weaknesses.

---

> ### Author Response · Authors · 2025-11-20
> **Generalizable Framework Diagram**
>
> Q. Overly Simplified and Non-Generalizable Framework Diagram (Fig. 3): The main diagram is a simple linear flow that fails to illustrate core algorithmic details. More importantly, it is Soybean-specific (showing trifoliate synthesis) and completely omits the process for the second example crop, tomato, undermining the claim of a general "framework."
>
> Response. Thank you for your invaluable suggestion. The revised manuscript now includes an updated and more generalized framework block diagram. The plant-level component has been expanded to illustrate the spiral phyllotactic generator, providing a clearer representation of foliage structure synthesis. Additionally, a configuration file block has been incorporated to indicate the configurable parameters and their correspondence framework components.

---

> ### Author Response · Authors · 2025-11-20
> **Comparison with random copy-pasting**
>
> Q. Lack of Simple Synthesis Baseline Comparison: A key ablation is missing: a comparison against a much simpler baseline, such as random copy-pasting of leaves. Without this, it is unclear if the complex biological modeling (e.g., phyllotaxis) is truly necessary for effective transfer learning.
>
> The result of random copy-pasting is provided in an Ablation study in Appendix A.5, p. 17 with random copy-pasting with significantly less accuracy. For soybeans with 15% diseased leaves, as an example, the results (based on accuracy) for our method and random copy pasting illustrate the effectiveness of our method, as follows. DenseNet achieved the highest baseline accuracy of 91.47%, which dropped to 84.66% under random copy–paste augmentation. Similarly, ResNet and CBAM also experienced declines from 85.31% down to 82.01% and from 77.76% down to 73.15%, respectively. The Swin Transformer model, being spatially sensitive, showed compatibly significant performance reduction (72.36% down to 65.94%). The results clearly indicate that the introduction of random copy–paste augmentation leads to a notable decline in accuracy across all architectures, confirming that random placement of diseased regions disrupts the natural spatial correlation between disease patterns and the leaf structure.

---

> ### Author Response · Authors · 2025-11-20
> **Ambiguity in formula**
>
> Q. Ambiguity and Errors in Core Mathematical Formulation (Sec. 3.2): The formulas for leaf positioning contain two issues: (a) The golden angle definition () conflicts with the use of the non-standard in the cosine/sine functions, creating a unit ambiguity. (b) An "off-by-one" error exists in the index range, which does not match the intended leaf count N.
>
> Response. Thank you very much for pointing out that ambiguity. The revised manuscript has been refined accordingly, and your identified ambiguities have been thoroughly clarified to improve clarity and readability.

---

> ### Author Response · Authors · 2025-11-20
> **Comparison of Distribution with real-world data**
>
> Q. Lack of Real-World Distribution Validation for Disease Modeling: The paper claims its disease modeling (Fig. 4) is based on real data, but only presents statistics for the synthesized dataset. To justify fidelity, the authors must first present an analysis of the real-world target dataset (MH-SoyaHealth Vision) to show that the distribution characteristics are faithfully replicated.
>
> Response. Thank you for your invaluable feedback. Our proposed method is grounded in observations of disease distribution patterns in real-world foliage images from the MH-SoyaHealthVision dataset, as well as insights from prior studies on the considered plants (see refereneces below). The revised manuscript has been updated to include the disease distribution analysis of real-world diseased foliage images (see Figure 4, p. 8), providing empirical support for our approach, per your comment.
>
> [1] Jiemeng Tao, Peijian Cao, Yansong Xiao, Zhenhua Wang, Zhihua Huang, Jingjing Jin, Yongjun Liu, Huaqun Yin, Tianbo Liu, and Zhicheng Zhou. Distribution of the potential pathogenic alternaria on plant leaves determines foliar fungal communities around the disease spot. Environmental Research, 200:111715, 2021. ISSN 0013-9351. doi: https://doi.org/10.1016/j.envres. 2021.111715.
>
> [2] XB Yang, JP Snow, and GT Berggren. Patterns of rhizoctonia foliar blight on soybean and the effect of aggregation of disease development. Phytopathology, 81(3):287–293, 1991.

---

> > ### Comment · Reviewer_4gCG · 2025-11-27
> > **maintain my score**
> >
> > I choose to maintain my score.

---

### Official Review · Reviewer_j62w · 2025-11-03

**Soundness:** 2
**Presentation:** 2
**Contribution:** 2
**Rating:** 4
**Confidence:** 3

**Summary:**

This paper discusses a flaw in many prior literature -- where they do crop disease classification by taking close-up shots of leaf. They rather discuss how UAVs capturing foliage are more practical. They discuss the lack of datasets for foliage-based crop disease classification. The authors introduce a framework called foliagen -- where they combine leaf images from multiple existing datasets to create foliage imagery. These datasets (considered annotated) were used for downstream tasks and showed performance improvement.

**Strengths:**

+ The paper is easy to follow
+ The practicality of UAV foliage over leaf imagery is reasonable
+ The proposed method seems simple (and thus attractive!)
+ The performance improvement on the downstream tasks are encouraging

**Weaknesses:**

- I like the authors' motivation of UAV imagery being more practical. How does UAV imagery actually look like? I think even a small dataset of actual imagery would be very useful -- it would show various real-world challenges (shading, wind, etc.)

- Is it merely enough to combine leaf disease images. Would diseases not show up in other parts of the plant?

- I think knowing related work on generative plant disease classification/datasets would be useful. Currently, this seems to be a geometric method of combining leaves. Are there pixel-level methods?

**Questions:**

- As mentioned in the weakness, I would like to see the justification of this imagery being representative of actual imagery collected via UAVs

- It would be good to see justification of absence of disease artifacts in other parts of the plant.

- Is it possible to view results on additional plants available in the existing datasets.

- Is there any prior work on generative plant disease classification/datasets? If so, could you contrast?

---

> ### Author Response · Authors · 2025-11-20
> **Comparison of UAV foliage image and generated foliage image**
>
> Q. I would like to see the justification of this imagery being representative of actual imagery collected via UAVs
>
> Response. Foliagen is designed to emulate real-world foliage imagery captured by UAVs, where most of the scene is densely covered with leaves exhibiting varying numbers of disease hotspots depending on the disease stages. To demonstrate this realism, the revised manuscript now includes an example of a real-world foliage image (see Appendix A.2). Furthermore, the disease distribution analysis of real-world foliage images has been incorporated into the main text on p. 8, illustrating a strong similarity between the generated and natural distributions (see Figure 4).

---

> > ### Comment · Reviewer_j62w · 2025-11-27
> >
> > Thanks. This is useful. Appendix A2 images are useful; Figure 4 KDEs look a bit different; in 4b there is no mass beyond 2, whereas in 4c there is.
> >
> > Can you clarify this please?
> >
> > Another way to compare would be to draw them on the same plot.

---

> > > ### Author Response · Authors · 2025-11-27
> > >
> > > Q. Figure 4 KDEs look a bit different; in 4b there is no mass beyond 2, whereas in 4c there is. Can you clarify this please?
> > >
> > > Response. Thank you very much for your constructive feedback.
> > > Most natural foliage exhibits one or two disease hotspots, although a small number may contain fewer or more hotspots. Foliagen takes this distribution into account by modeling only one or two hotspots, thereby prioritizing the majority case. Consequently, Figure 4b does not show mass beyond two hotspots, as the framework focuses on the most prevalent hotspot patterns observed in real foliage.
> > > In future work, we will adapt the hotspot assignment algorithm so that the synthesized foliage can more closely mimic the full natural variability of disease hotspot occurrence.

---

> ### Author Response · Authors · 2025-11-20
> **Justification of absence of disease artifacts in other parts**
>
> Q. It would be good to see justification of absence of disease artifacts in other parts of the plant.
>
> Response. Appendix A.2 presents UAV-captured foliage images of soybean. Observations of these images indicate that disease symptoms are typically confined to limited regions, while the majority of the foliage remains healthy. Reflecting this natural pattern, Foliagen similarly maintains predominantly healthy leaf areas, with diseased leaves appearing only in localized regions, thereby ensuring a realistic spatial distribution of disease. Also below listed studies supports the pattern.
>
> [1] Jiemeng Tao, Peijian Cao, Yansong Xiao, Zhenhua Wang, Zhihua Huang, Jingjing Jin, Yongjun Liu, Huaqun Yin, Tianbo Liu, and Zhicheng Zhou. Distribution of the potential pathogenic alternaria on plant leaves determines foliar fungal communities around the disease spot. Environmental Research, 200:111715, 2021. ISSN 0013-9351. doi: https://doi.org/10.1016/j.envres. 2021.111715.
>
> [2] XB Yang, JP Snow, and GT Berggren. Patterns of rhizoctonia foliar blight on soybean and the effect of aggregation of disease development. Phytopathology, 81(3):287–293, 1991.

---

> ### Author Response · Authors · 2025-11-20
> **Is it possible to view results on additional plants available in the existing datasets.**
>
> Q. Is it possible to view results on additional plants available in the existing datasets.
>
> Response. We have attempted to obtain leaf images of other crops, including peanut plant.  Our current effort lies in taking and properly annotating quality images of diseased and healthy single peanut leaves and peanut foliages. Given its highly time-consuming effort to obtain the peanut leaf image dataset, however, we are presently constrained from expanding our analysis to include additional plants.

---

> ### Author Response · Authors · 2025-11-20
> **Prior works**
>
> Q. Is there any prior work on generative plant disease classification/datasets? If so, could you contrast?
>
> Response. Yes, there are existing studies on generative approaches for plant disease classification; however, the majority of those studies focus exclusively on single-leaf disease generation and classification. To the best of our knowledge, Foliagen is the first framework to address generative foliage-level disease classification, capturing the complexity of multiple leaves within a natural canopy context. The revised manuscript now expands the Related Work section to include discussions on prior generative data synthesis methods. The following newly cited papers have been added to our revised manuscript so as to provide comprehensive context:
>
> [1] Amreen Abbas, Sweta Jain, Mahesh Gour, and Swetha Vankudothu. Tomato plant disease detection using transfer learning with c-gan synthetic images. Computers and Electronics in Agriculture, 187:106279, 2021.
>
> [2] Quan Huu Cap, Hiroyuki Uga, Satoshi Kagiwada, and Hitoshi Iyatomi. Leafgan: An effective data augmentation method for practical plant disease diagnosis. arXiv preprint arXiv:2002.10100, 2020.
>
> [3] Omkar Khare, Sachin Mane, Harish Kulkarni, et al. Leafnst: an improved data augmentation method for classification of plant disease using object-based neural style transfer. Discov Artif Intell, 4:50, 2024.
>
> [4] Alessandro Benfenati, Davide Bolzi, Paola Causin, and Roberto Oberti. A deep learning generative model approach for image synthesis of plant leaves. Plos one, 17(11): e0276972, 2022.

---

> ### Author Response · Authors · 2025-11-20
> **Would disease not show up in other parts of the plant?**
>
> Q. Would disease not show up in other parts of the plant?
>
> Response. While plant diseases may manifest in roots, stems, or fruits, the scope of our work is confined to symptoms that are visually observable on leaves.

---

> > ### Comment · Reviewer_j62w · 2025-11-27
> >
> > >Q. Would disease not show up in other parts of the plant?
> >
> > >Response. While plant diseases may manifest in roots, stems, or fruits, the scope of our work is confined to symptoms that are visually observable on leaves.
> >
> >
> > My main issue with this is -- while you use the leaves primarily; but not changing the other parts of a diseased plant may lead to visual differences.

---

> ### Author Response · Authors · 2025-11-20
> **Are there pixel-level methods?**
>
> Q. Are there pixel-level methods?
>
> Response. Yes, there are pixel-level generative methods, such as PixelCNN. We experimented with the PixelCNN model using the MH-SoyaHealthVision dataset to generate foliage images. However, the resulting outputs lacked structural coherence and failed to resemble realistic foliage. A representative sample of such generated images is included in Appendix A.2.1 for information.

---

> ### Author Response · Authors · 2025-11-27
>
> Q. My main issue with this is -- while you use the leaves primarily; but not changing the other parts of a diseased plant may lead to visual differences.
>
> Response. Thank you for your constructive feedback.
> The symptoms of predominant soybean diseases show up just in leaves even at disease onsets, considered to be most critical points of time for automated disease detection and identification for effective disease treatment. Such symptoms are observable from the bird-eye view (as taken by UAVs or tractor-mounted cameras). Our Foliagen intends to generate foliage images for training and evaluating ML models in order to permit the best choice of suitable model and future model development for real-world applications. At the disease onset, little changes happen to the other parts of a diseased plant (like its stems).
>
> A couple of disease categories, e.g., Root/Stem Rot, caused by fungi on plant stems, particularly near the soil line.  They don't manifest themselves in leaf symptoms at the onset stage. However, they cause affected plants' leaves to wilt before plants' collapses at advanced disease stages, when they can be detected (but no effective treatment may be administrated). Our Foliagen may produce foliage images under such disease categories as well, by having yellow leaves at the plant level for one or two patches per image.

---

### Official Review · Reviewer_fBth · 2025-11-05

**Soundness:** 3
**Presentation:** 3
**Contribution:** 3
**Rating:** 8
**Confidence:** 4

**Summary:**

This paper proposes Foliagen, a framework for generating synthetic crop foliage datasets from existing single-leaf images. The method bridges the gap between single-leaf data and real UAV canopy imagery by simulating multi-plant scenes with realistic spatial structure and disease distribution. Foliagen builds images through a three-level process (leaf, plant, foliage) and controls disease severity by leveraging a controllable parameter.

Datasets for soybean and tomato are synthesized using public leaf images (ASDID, Kaggle, PlantVillage), with leaves arranged naturally on soil backgrounds to resemble real plant growth. Five SOTA classifiers are evaluated, showing that models pretrained on Foliagen data generalize better to real canopy datasets (e.g., MH-SoyaHealthVision), especially under limited real data. The work establishes a practical benchmark for studying leaf-to-foliage generalization and developing field-ready crop disease models.

**Strengths:**

(1) The paper addresses a highly meaningful and underexplored problem: transferring disease recognition knowledge from single-leaf images to UAV-level canopy imagery. This leaf-to-foliage generalization idea is both novel and practical, as it provides a concrete pathway to bridge laboratory datasets and real field conditions. To the best of my knowledge, very few studies have systematically investigated this level shift, making Foliagen an important step toward realistic and scalable agricultural vision benchmarks.

(2) The proposed Foliagen framework enables controllable and biologically inspired data generation. It introduces a hierarchical process from leaf to plant to foliage with a tunable disease ratio parameter gamma, allowing users to precisely control the severity and spatial distribution of disease symptoms. In addition, the use of spiral phyllotaxis, a natural leaf arrangement pattern following the golden angle of 137.5 degrees, makes the synthetic foliage geometrically realistic and closer to actual crop canopies. This balance between controllability and biological plausibility distinguishes Foliagen from conventional rule-based augmentation pipelines.

(3) The paper demonstrates the practical value of Foliagen through transfer learning experiments. Models pretrained on the synthetic foliage data achieve higher accuracy on real UAV canopy images, even when fine-tuned with only a small fraction of real samples. This shows that the generated data effectively supports real-world applications and can reduce the reliance on costly field annotations.

**Weaknesses:**

(1) The data generation process in Foliagen is primarily rule-based and does not involve any learnable modeling. While the proposed pipeline effectively demonstrates controllable canopy synthesis, incorporating modern generative methods such as GANs or diffusion models could further improve realism and variability. Even if such approaches may not always outperform rule-based designs, exploring them would offer valuable insights into the trade-off between generative plausibility and diversity.

(2) The paper does not include ablation or sensitivity analysis to evaluate the contribution of each generation level. Introducing more controlled variations across the leaf, plant, and foliage stages could help quantify their individual effects and justify the design choices in Foliagen.

**Questions:**

(1) It would be useful to introduce more variations at different generation levels (leaf, plant, and foliage) and analyze how these affect model performance. Such experiments could help clarify the role of each level in shaping the learned representations and validate the effectiveness of the hierarchical design.

(2) A comparison with generative approaches, such as GANs or diffusion models, could be considered to assess the trade-off between controllability and realism. Such an analysis would provide deeper insight into how rule-based synthesis compares with learnable generation in modeling complex canopy structures.

(3) The sensitivity of the results to the choice of disease ratio (e.g., 5% vs. 15%) should be more thoroughly analyzed, as it likely affects model training and transferability. It would also be valuable to discuss how the ratio can be set to better reflect real-world field conditions, for example, by relating it to typical disease incidence observed in UAV imagery.

---

> ### Author Response · Authors · 2025-11-20
> **Comparison with generative approaches**
>
> Q. A comparison with generative approaches, such as GANs or diffusion models, could be considered to assess the trade-off between controllability and realism. Such an analysis would provide deeper insight into how rule-based synthesis compares with learnable generation in modeling complex canopy structures.
>
> Response. Thank you for the insightful recommendation. We experimented with the Deep Convolutional Generative Adversarial Network (DCGAN) and Denoising Diffusion Probabilistic Model (DDPM) using the MH-SoyaHealthVision dataset to generate synthetic foliage images. The generated outputs, as might be expected, lacked realism and failed to accurately depict leaf disease symptoms. While the diffusion-based model produced visually plausible foliage images, it still did not capture the distinct pathological features of diseased leaves. Therefore, a rule-based synthesis approach, like our Foliagen, proved more effective for generating realistic and symptom-preserving foliage images. Sample outputs produced by both DCGAN and DDPM are included and discussed in Appendix A.2.1.

---

> ### Author Response · Authors · 2025-11-20
> **Ablation study**
>
> Q.  It would be useful to introduce more variations at different generation levels (leaf, plant, and foliage) and analyze how these affect model performance. Such experiments could help clarify the role of each level in shaping the learned representations and validate the effectiveness of the hierarchical design.
>
> Response. Thank you for your suggestion. We experimented with the framework by removing the structured arrangement of leaves (Case 1) and retaining background (Case 2) of the image as our ablation study. Note that all of the generated images are for a disease rate of 15%. In both cases, the accuracy for all models decreased, with Case 2 displaying significant performance degradation. The results have been included in the revised manuscript (see Appendix A.5 on p. 17).
>
> Example results for soybean are given below:
>
> Model      | Ours   | without arrangement | With Background
>
> ----------------------------------------------------------------------------
>
> VGG         76.26    	| 69.26      		| 37.52
>
> Densenet   91.47    	| 84.66      		| 76.66
>
> CBAM       77.76    	| 73.15     		| 49.15
>
> Swin          72.36    	| 65.94      		| 32.58
>
> ResNet      85.31    	| 82.01      		| 45.78

---

> ### Author Response · Authors · 2025-11-20
> **Sensitivity to disease rate**
>
> Q.  The sensitivity of the results to the choice of disease ratio (e.g., 5% vs. 15%) should be more thoroughly analyzed, as it likely affects model training and transferability. It would also be valuable to discuss how the ratio can be set to better reflect real-world field conditions, for example, by relating it to typical disease incidence observed in UAV imagery.
>
> Response. Figure 10 (on p. 16) illustrates the sensitivity analysis of soybean under varying disease rates, concluding that model’s performance (based on accuracy) changes significantly as the severity or proportion of disease increases. DenseNet exhibits the greatest stability, maintaining high performance even at low disease rates, increasing only moderately from 88.64% at 2% disease rate to 96.83% at 50%, a small gain of 8.19%. ResNet also performs reliably across disease levels, rising from 79.24% to 89.45% with a moderate gain of 10.21%. In contrast, CBAM is highly sensitive to the disease rate, starting with a lower accuracy of 63.52% at 2% but sharply improving to 90.47% at 50%, representing the highest gain of 26.95% among all models. Likewise, VGG and Swin also show strong dependence on higher disease rates: VGG increases from 65.24% to 85.56% (20.32%), and Swin from 64.01% to 84.18% (20.17%).
>
> Our proposed method is grounded in observations of disease distribution patterns in real-world foliage images from the MH-SoyaHealthVision dataset, as well as insights from prior studies on the considered plants. The revised manuscript has been updated to include the disease distribution analysis of real-world diseased foliage images (see Figure 4 on p. 8), which illustrates strong similarity of disease distribution, providing empirical support for our approach, per your comment.

---

> > ### Comment · Reviewer_fBth · 2025-11-27
> >
> > The authors’ responses and revisions address my questions. I have no further concerns and will keep my original score (8).

---

### Meta-Review · Area_Chair_zZzk · 2025-12-29

**Summary:**

# Decision

Reviewer opinions were mixed, with a slight overall leaning toward acceptance. However, this inclination appears to be influenced in part by a relatively superficial assessment of the submission by one reviewer. While the paper addresses a relevant problem and proposes a conceptually simple, biologically inspired procedural framework, the submission falls short of convincingly supporting its core claims.

The approach is largely rule-based and relies on strong but insufficiently-articulated assumptions that limit its generality, with several key methodological details missing (e.g., leaf normalization, plant assembly, and opt. realism considerations). More critically, the evaluation protocol does not appear well aligned with the stated objectives. The reported results rely mainly on fine-tuning experiments and do not include more direct and informative tests, e.g., by training purely on synthetic data and evaluating directly on real imagery.

Overall, despite positive aspects in terms of motivation and informed design of this procedural framework, the methodological gaps and unconvincing evaluation prevent a clear and rigorous assessment of the contribution. More thorough clarification and more targeted experiments would be necessary for the work to meet the bar for acceptance.

------------
# Justification

## Consolidated Remarks from Reviewers

### Strengths

#### Lean solution targeting a meaningful problem
- Meaningful problem tackled (transferring disease recognition knowledge from single-leaf images to UAV-level canopy imagery) [`fBth`, `j62w`, `4gCG`]
- Procedural generative framework inspired from plant biology [`fBth`, `4gCG`]
- Parameterizable process [`fBth`]
- Simple (thus attractive) process [`j62w`]

#### Evaluation via transfer learning
- Experimental validation via transfer learning (ML models pre-trained on the proposed synthetic dataset and tuned on real data performed better than the same models directly trained on the real images) [`fBth`]
- Ablation study of some biology-inspired parameters [`4gCG`]

#### Misc.
- Easy to read [`j62w`]


### Weaknesses

#### Non-generalizable framework
- Purely rule-based generation, with no learnable components [`fBth`]
- Overly simplified, non-generalizable framework [`4gCG`]
- Lack of comparison to actual UAV imagery [`j62w`]
- Lack of comparison to real-data distribution (e.g., in terms of disease modeling) [`4gCG`]
- Solution only considering the impact of disease on leaf appearance and not on other aspects of the plants [`j62w`]
- A few notation errors [`4gCG`]

#### Partial evaluation and discussions
- No comparison to other baselines (learning-based, pixel-wise, etc.) [`fBth`, `j62w`]
- Minimal ablation w.r.t. generation levels and their variations [`fBth`, `4gCG`]
- No analysis w.r.t. impact of procedural parameters (background, arrangement, etc.) [`fBth`]
- Lack of discussion w.r.t. choice of disease ratio and its impact on model performance [`fBth`]


## AC's Own Concerns

### Poor methodological formalism
- The authors extract single leaves from existing datasets but do not explain how these leaves are registered into a common coordinate system before being assembled into plants. E.g., variations in orientation, scale, and pose across images could significantly affect realism, yet this issue is neither addressed nor discussed. The authors might assume that all single-leaf images are pre-registered to the same coordinate system (which might be the case, e.g., with ASDID), but this assumption and its impact on the data collection should be properly stated and discussed.
- Similarly, other sources of visual heterogeneity among single-leaf images (e.g., camera exposure, resolution, illumination) are not considered.
- Several assumptions underlying the plant assembly process are left implicit. For instance, while the prely-2D placement scheme may be justified by the UAV (bird’s-eye) viewpoint, the lack of distance-based scaling, depth cues, occlusion, or shadow handling is not discussed.
- While the authors claim that their framework can generalize to other crops, their algorithm only models plants with similar structure (spiral phyllotaxis).

### Unconvincing evaluations

- Key experiments in Table 1 (models trained and tested on real data) and Table 3 (models trained and tested solely on synthetic data) do not support the paper’s central claim, as they provide no insight into the usefulness or realism of the proposed generative process.

- The main supporting evidence comes from Table 2, where models are pre-trained on synthetic data and fine-tuned on real data. While suggestive, this setup only indirectly evaluates the proposed framework.

- The most direct and informative evaluation, i.e., training models purely on synthetic data and testing them directly on real data without fine-tuning, is missing. Overall, the experimental design is not well aligned with the stated goal of validating the effectiveness of the proposed data generation scheme.

- While the ablation study in the sup-mat is somewhat informative, the others experiments there are also unconvincing.
Table 4 does not isolate the benefit of the proposed synthetic data, as it compares models trained under the same regime (pretraining on single-leaf dataset then finetuning on proposed dataset) rather than across different training data configurations (e.g., with or without the finetuning on proposed data).
Finally, although the authors attempted to include learning-based generative baselines (e.g., DCGAN, PixelCNN) c.f. reviewer `fBth`'s suggestion, these models clearly failed to converge (Figure 9). With appropriate training data and configurations, modern generative approaches (GANs or diffusion models) should be capable of producing at least moderately realistic imagery. This leaves the authors' attempt inconclusive.

**Reviewer Concerns:**

See above for summary of main concerns shared by reviewers.

The authors clarified a few misunderstandings (e.g., presence of some ablation study in the sup-mat) and provided some justifications (e.g., comparison to real-world data distribution, non-visibility of disease symptoms on other parts of the plants in UAV imagery, insight in disease rate, etc.).

However, their experiments (e.g., comparison to learning-based solutions) still fall short of demonstrating the effectiveness and realism of their framework.

**Reviewer Scores:**

### Reviewer `fBth`
- **Original score:** 8
- **Score change:** maintained their score, c.f. own reply.

### Reviewer `j62w`
- **Original score:** 4
- **Score change:** likely kept their score, c.f. additional issues raised during exchange with authors.

### Reviewer `4gCG`
- **Original score:** 6
- **Score change:** maintained their score, c.f. own reply.

---

### Decision · Program_Chairs · 2026-01-26

Reject